# The Post-Invasion Population Dynamics and Damage Caused by Globose Scale in Central Eurasia: Destiny of Wild Apricot Still at Stake

**DOI:** 10.3390/insects16040409

**Published:** 2025-04-13

**Authors:** Ping Zhang, Yifan Li, Cuihong Li, Guizhen Gao, Zhaoke Dong, Elahe Rostami, Zhaozhi Lu, Myron P. Zalucki

**Affiliations:** 1State Key Laboratory of Desert and Oasis Ecology, Xinjiang Institute of Ecology and Geography, Chinese Academy of Sciences, Urumqi 830011, China; zpsds2018@163.com; 2The Specimen Museum of Xinjiang Institute of Ecology and Geography, Chinese Academy of Sciences, Urumqi 830011, China; 3University of Chinese Academy of Sciences, Beijing 100049, China; 4Shandong Engineering Research Center for Environment-Friendly Agricultural Pest Management, College of Plant Health and Medicine, Qingdao Agricultural University, Qingdao 266109, Chinazhaoke_dong@126.com (Z.D.); 5College of Forestry and Landscape Architecture, Xinjiang Agricultural University, Urumqi 830052, Chinagaoguizhen1984@163.com (G.G.); 6Fujian Provincial Key Laboratory of Quality and Safety of Agricultural Products, Institute of Quality Standards & Testing Technology for Agro-Products, Fujian Academy of Agricultural Sciences, Fuzhou 350003, China; elaherostami@163.com; 7School of the Environment, The University of Queensland, St. Lucia, QLD 4072, Australia

**Keywords:** *Armeniaca vulgaris*, fruit set, plant compensation, *Sphaerolecanium prunastri*, wild apricot forest

## Abstract

*Sphaerolecanium prunastri* (Boyer de Fonscolombe) is a major invasive pest in the remnant wild apricot forests of the Tianshan Mountains in Xinjiang, China. To clarify its post-invasion population dynamics and damage in wild apricot forests, we surveyed infested and uninfested sites from 2019 to 2024. The results showed that damage is declining; however, the survival of wild apricots is still threatened by persistent high densities of *S. prunastri*. In addition, the cumulative damage from the scale has significantly reduced the trees’ reproductive capacity. It is essential to strengthen the monitoring and integrated pest management (IPM) strategies against the scale to reduce its density below damaging levels.

## 1. Introduction

Scale insects are common and destructive pests that frequently infest horticultural, fruit, and ornamental plants [1]. Their small size makes them difficult to detect and identify, facilitating their rapid spread to new areas [2,3]. Some scale species can cause significant yield losses, economic damage, and even profound ecological impacts in natural systems. For example, the invasion of tortoise pine scale (*Toumeyella parvicornis* Cockerell) in Turks and Caicos Islands has resulted in the infestation of more than 90% of the pine forested area and severe damage to several *Pinus* species, seriously disturbing the balance of the forest ecosystem [4]. This species has also invaded Europe [5], causing higher mortality of stone pines (*Pinus pinea* L.) in a coastal pinewood in Southern Italy [6]. Another species, known as roseau cane scale (*Nipponaclerda biwakoensis* Kuwana), originates from China and Japan [7]. Since its discovery in the Mississippi River Delta of the United States, it has caused extensive die-offs and the decline of roseau canes, severely disrupting the balance of the marsh ecosystem [8,9].

The globose scale (GS) *Sphaerolecanium prunastri* (Boyer de Fonscolombe) (Hemiptera: Coccoidea), a soft scale insect that is an invasive species in China [10]. Native to central and southern Europe, it has now spread across the Mediterranean, North America, the Middle East, and Central and East Asia [11], with records from Spain, France, Italy, Belgium, the United States, Turkey, South Korea, Uzbekistan, Turkmenistan, and China [12]. GS is a key pest of *Prunus* spp. [13], severely damaging plum, peach, and apricot trees [12,14,15], with infestation rates reaching 100% in some orchards [13,16]. In China, this pest has been recorded in Liaoning, Shandong, Hebei, and Shaanxi provinces [10]. However, there are no detailed reports on the extent of the damage in these regions. In 2019, an outbreak of GS was detected in wild apricot (*Armeniaca vulgaris* Lamarck.) forests in Xinjiang, China [10], particularly in the Ili Valley of the Tianshan Mountains, which has 1440 ha of patchily distributed remnant wild apricot forests [17]. The invasion pathway of GS into Xinjiang remains undetermined. While human-mediated dispersal may be a potential contributing factor, the presumption nevertheless requires further validation through comprehensive sampling and genetic analyses. The GS infestation has led to a decline in the physiological health of wild apricot populations, causing dieback, stunted growth, and increased mortality of trees. As the ancestor of the cultivated apricot, this tertiary relict plant serves as a vital genetic resource for further breeding [18]. It is estimated that more than 80% of wild apricot trees in this valley have been infested with the pest [19], posing a serious threat to these trees and the resilience of the wild apricot forest ecosystem.

GS has one generation a year and overwinters as 2^nd^-instar nymphs on wild apricot trees [10]. The 2^nd^-instar nymph goes through the longest life history stage. At the beginning of July, 2^nd^-instar nymphs feed on the phloem of young branches and then overwinter from October to late March of the following year. After overwintering, pupation occurs from late March to early April [10]. The 2^nd^-instar nymph stage coincides with bud development and flowering of wild apricots. The long-term damage stress from GS seriously affects the flowering and germination of wild apricots [20]. Adults appear in late May and early June, when their feeding results in a large amount of honeydew [10]. Eggs hatch in early to mid-June and after a short period of crawling, 1^st^-instar nymphs settle on young branches (less than 3 years old) and feed on the phloem, developing into 2^nd^-instar nymphs in July [10]. Infested wild apricots are seriously damaged by female adults and 1^st^-instar nymphs in early summer, which coincides with the most vigorous growth and fruit-setting period [21].

The sap-sucking behavior of scale insects causes a cascade of direct effects, including branch dieback and secondary damage such as sooty mold [22], which reduces photosynthetic activity and stunts the growth of infested branches [18]. To mitigate the impact of GS on wild apricots, various management strategies have been implemented. The introduction of ladybird beetles (e.g., *Harmonia axyridis* Pallas) as a biological control agent had limited success, with a field efficacy below 35% [23]. Although a variety of natural parasitic enemies have been identified within wild apricot forests [24,25], their impact on GS populations is low [26]. Since 2022, a large-scale aerial spraying program has been initiated to suppress GS populations. Nevertheless, its control efficacy has been challenging due to the mountainous topography, adverse weather, and GS biological traits such as a heavy protective waxy covering. As a result, GS populations continue to persist at damaging levels and pose a major risk to the germplasm of wild apricots.

A rational assessment of pest damage can provide the foundation for decisions on management, especially in targeting areas to spray, selecting appropriate methods, and determining the optimal timing of controls [27]. There are some basic assessments for wild apricot damage, including GS density and the damage caused during initial GS outbreaks [19], but there remains a gap between the invasion, rapid spread, and the practical requirements for management of GS, which has a more extensive and complex impact on wild apricots over time. This includes changes in the damage caused by GS and the cumulative effects on the growth and reproductive capacity of wild apricots. Such assessments are needed to monitor both pest population dynamics and tree heath simultaneously over a long period.

In this study, we clarify the status of the GS population and the extent of the damage since its initial outbreak. Specifically, we (1) survey population changes of GS by estimating pest density over 6 years following its first detection; (2) assess the change in GS damage to wild apricot trees over this period; and (3) evaluate the cumulative effects of GS damage on tree growth and reproduction, including spring shoot growth, fruit yield, and flower bud density.

## 2. Materials and Methods

### 2.1. Experimental Design

#### 2.1.1. Survey Site

This study was carried out from 2019 to 2024 (excluding 2023) in wild apricot forests located in Gongliu, Xinyuan, and Huocheng Counties, China, at latitudes of 43°14′02″ N to 44°42′88″ N, longitudes of 83°03′87″ E to 83°48′54″ E, and 1026.33–1376.6 m above sea level.

Four areas in Gongliu County and two areas in Xinyuan County, comprising 90% of the distribution of remnant forests where wild apricots are the dominant species and were severely infested by GS, were selected as experimental sites (Figure 1). Another area, Huocheng County, was the control site, as wild apricots were not yet infested by GS. At each survey location, we established replicate circular sample plots with a radius of 20 m for a total 12, 4, and 3 plots in Gongliu, Xinyuan, and Huocheng Counties, respectively. Replicate plots in each site were spaced 1–3 km apart (Table 1).

#### 2.1.2. Density Survey of *Sphaerolecanium prunastri*

GS density surveys were carried out from 2019 to 2024, except for 2023. Surveys of nymph density began in early April, after the overwintering phase of the 2^nd^-instar (Figure 2a). The density of 1^st^-instar nymphs (Figure 2b) was estimated in early to mid-June. In each infested area, we randomly sampled 15 trees with different damage rankings located in the central areas to assess nymph density. The randomized sampling followed a zigzag pattern in all plots and sites. A 3-year-old branch (1.5–3 cm in diameter) was randomly selected from the east, south, west, and north directions of each tree at a height of 4–6 m and cut down. Branchlets measuring 20 cm in length were used to count 1^st^- and 2^nd^-instar nymphs because GS was concentrated on these branches. All densities reported in the results are per 20 cm by the same observer to maintain consistency.

#### 2.1.3. Damage Ranking of Wild Apricots

To assess the damage due to GS infestation, we surveyed the damage rankings of trees and the number of dead individuals. Tree damage was categorized as: 0 = non-infested, 1 = least damaged (1–10%), 2 = low damage (11–30%), 3 = medium (31–50%), 4 = high (51–70%) and 5 = very high (71–100%) damage based on the percentage of branches damaged by GS in each tree (Figure 3). This monitoring was conducted in early to mid-June between 2021 and 2024 by PZ for consistency. We assessed the mortality of wild apricots in each plot to evaluate the cumulative effect of GS on tree health in 2021 and 2024. The dead individuals were identified based on the presence of old GS damage and the absence of new branches and green leaves.

#### 2.1.4. The Growth of Wild Apricots

The growth of spring shoots can indirectly reflect the potential growth and vigor of trees within a year. We observed the length of spring shoots (fresh shoots of the current year) on branches across trees with different damage rankings from 2021 to 2023. In early to mid-June, when wild apricot tree growth is at the most vigorous stage, 15 trees were randomly chosen from non-infested, low, and high damage rankings and labelled. Non-infested trees, serving as controls, were chosen from all plots in Huocheng County, which were free from pest scale. There were no pest-free trees in infested counties. The low- and high-damage trees were selected from plots in Gongliu and Xinyuan counties. Following the method used for surveying the densities of 1st-instar nymphs, we cut four 3-year-old branches (less than 3 year growth) and measured the length of five spring shoots on each branch using a tape measure. A total of 20 spring shoots were measured per tree.

#### 2.1.5. Assessment of Flower Buds and Fruit Yield

In 2020, we assessed the density of flower buds at three survey sites in Gongliu County. Then, from 2022 to 2024, we systematically assessed fruit yield within all plots and assessed flowering in 2023.

The assessment of flower buds was carried out in plots located in the Xiaomohuer, Jinqikesai and Kuolesai Valleys (see Table 1). Five wild apricot trees were randomly selected from low, medium, and high damage rankings in each plot. In late August and early September, one 3-year-old branch was cut in the east, south, west, and north directions of each tree. The length of the branch was measured and the number of flower buds counted [28,29], and the density of flower buds per meter was calculated.

Assessment of fruit yield and flowering: In late June to early July, during the ripening of wild apricot fruit, we estimated the fruit yield for all non-infested trees and those with low and high levels of damage by counting the number of fruits, as weighing all fruit was impractical. Non-infested trees were sampled from three plots in Huocheng County, while low- and high-damage trees were selected from all plots in Gongliu and Xinyuan Counties. To establish a weight estimation criterion, we weighed 50 fruits (approximately 500 g) in each plot, which allowed us to estimate the yield for each plot. Furthermore, in early to mid-April 2023, when wild apricots are in full bloom, we assessed the flowering rate of low-, medium-, and high-damage trees across all plots in Gongliu County. The flowering rate was calculated as the percentage of blooms relative to the entire canopy (Figure 4).

### 2.2. Data Analysis

One-way analysis of variance (ANOVA) was used to analyze the differences in GS density, flower bud density, and percentage flowering across damage rankings (SPSS Statistics 20, IBMCorp, Armonk, NY, USA). Year and damage ranking were fixed factors, and the above observed variables were dependent variables. Data normality and homogeneity of variance were checked by Shapiro–Wilk tests. Two-way analysis of variance (ANOVA) was used to analyze the differences in spring shoot length and fruit yield across years and damage rankings. Year and damage ranking were fixed factors, survey site was a random factor, and other observed variables were dependent variables. The means were separated using Tukey’s post hoc test. Linear correlation (OriginPro. Version 2021, OriginLab Corporation, Northampton, MA, USA) was used to analyze the correlation between average damage ranking and fruit yield.

## 3. Results

### 3.1. Population Dynamics of GS Nymphs over Years

There was a significant difference among years for the densities of 1^st^- and 2^nd^-instar nymphs (1^st^: *F*_(4,34)_ = 13.63, *p* < 0.001; 2^nd^: *F*_(3,32)_ = 9.89, *p* < 0.001) (Figure 5). Both 1^st^- and 2nd-instar nymph densities exhibited a similar trend, rapidly increasing in 2020 and peaking in 2021 (1^st^: 1944 ± 325; 2^nd^: 454 ± 127). First-instar nymph density was low in 2019 at the start of the outbreak (38 ± 7) compared to other years. No significant differences were observed among 2020, 2021, 2022, and 2024. Second-instar nymph density was significantly lower in 2020 (137 ± 21) and 2024 (120 ± 19) compared to other years. The population densities of 2^nd^-instar nymphs declined to levels similar to 2020. However, 1^st^-instar nymph densities remained over 1000 per 20 cm.

### 3.2. Damage Rankings of Wild Apricots

There were significant differences in damage among years (Figure 6a). Between 2021 and 2023, while high and medium damage rankings increased to 29.6% and 27.6%, respectively, the high damage ranking decreased to 11.2% in 2024. Correspondingly, the low damage ranking increased from 56.2% to 67.3%, which indicated that the damage of wild apricot trees was gradually declining. However, the mortality of wild apricots in damaged areas was higher than that in non-infested areas. In 2021 and 2024, the percentages of dead trees in damaged areas reached 27.6% and 26.7%, while 13.0% and 12.4% dead trees occurred in non-infested areas in these years, respectively (Figure 6b).

### 3.3. Length of Spring Shoots Among Damage Rankings

GS infestation had a positive stimulating effect on the growth of spring shoots, but the effect was attenuated with time. The length of spring shoots was significantly related to year and damage ranking (*F*_(4,861)_ = 8.48, *p* < 0.001) (Figure 7). There was a positive correlation between the extent of damage and stem length. Non-infested trees exhibited stem lengths of 6.1 to 7.9 cm, while those with low and high damage had stem lengths of 7.9 to 18.3 cm and 15.1 to 20.7 cm, respectively. The length of spring shoots was 2–3 times longer on damaged trees compared to undamaged trees.

The shoot length response of infested trees gradually attenuated with time. The shoot lengths of trees with low and high damage rankings were significantly lower in 2022 and 2023 than in 2021. The shoot lengths of trees with low damage in descending order with year were 2021 (18.3 ± 0.4 cm) > 2022 (13.3 ± 10.8 cm) > 2023 (7.9 ± 0.4 cm). High-damage trees had stem lengths of 15.5 cm in 2023 and 15.1 cm in 2022, significantly shorter than the length of 20.7 cm in 2021 (Figure 7). However, the shoot growth of non-infested trees did not change significantly among the three years, indicating that GS had a cumulative damage effect on shoot growth. The length of the shoots gradually decreased with time (Figure 7).

### 3.4. Flower Buds and Flowering

GS negatively affected the reproductive capacity, affecting both flower buds and flower abundance per tree. The densities of flower buds and flowering percentages per tree were different among the three damage rankings (flower bud: *F*_(2,478)_ = 31.83, *p* < 0.001; flowering percentage per tree: *F*_(2,429)_ = 331.59, *p* < 0.001). With increasing damage, the flower bud density and flowering percentage per tree decreased. The flower bud density of low-damage trees (88.5 ± 4.7 per tree) was about 2 times higher than that of high-damage trees (46.2 ± 3.1) (Figure 8a). The flowering percentage of low-damage trees (59.8 ± 2.0 per tree) was more than 8 times higher than that of high-damage trees (7.5 ± 0.9 per tree) (Figure 8b).

### 3.5. Fruit Yield Loss

The fruit yield differed among the three damage rankings and years (*F*_(4,1241)_ = 30.58, *p* < 0.001) (Figure 9). The average fruit yield decreased with increasing damage ranking and year, except in 2024, which experienced extreme freezing conditions in the spring. In 2022 and 2023, the average fruit yield among the three damage rankings, in descending order, was non-infested (in 2022: 8.4 ± 0.9 kg; in 2023: 2.2 ± 0.4 kg) > low damage (2022: 2.0 ± 0.58 kg; 2023: 0.9 ± 0.2 kg) > high damage (2022: 0.3 ± 0.1 kg; 2023: 0.1 ± 0.1 kg). In 2024, the fruit yield of non-infested trees was less than 0.1 kg per tree and was not significantly different from uninfected trees. The fruit yield was reduced by more than 20 times in damaged trees compared to non-infested trees without considering the effect of freezing conditions on fruit yield in 2024. There was a significantly negative relationship between average fruit yield and damage ranking score (Figure 10), which indicated that GS had a decreasing effect on the fruiting of wild apricots.

## 4. Discussion

The invasion and subsequent outbreak of *S. prunastri* has had a profound detrimental impact on wild apricot forests, disrupting both individual tree growth and potentially the ecology of this unique system. Our observations revealed that GS remains at an outbreak level. Despite a gradual decrease in damage ranking within the wild apricot forests (Figure 6), heavily infested trees showed a counterintuitive response: longer spring shoots but significantly reduced fruiting and flowering. This suggests that the cumulative damage inflicted by GS stimulates compensatory branch growth while simultaneously suppressing reproductive capacity. Notably, this stimulatory effect abated over time (declining shoot growth, Figure 7), while uninfested trees remained unaffected.

### 4.1. GS Population Development and Damage in Wild Apricot Forests

GS females can establish a large population size in a short time because of their high fecundity, with 400–800 eggs per female [13,26]. The population of GS in our study areas developed rapidly in the two years following the initial detection of invasive populations in 2019. In wild apricot forests, the density of GS varies among regions. Wang et al. [19] showed that 1st-instar nymph density decreased from over 1000 to less than 500 in some areas from 2019 to 2020, while its density increased 100 times in others. From 2022 to 2024, the density and damage of GS began to decrease gradually. The decline was likely the result of large-scale aerial spraying and pruning management in 2022 and 2023. However, 1st-instar nymph density in 2024 was still 26 times higher than that in 2019. The cumulative damage from high GS density seems to accelerate the death of wild apricots. Compared to less than 13% mortality in uninfested areas, tree mortality nearly doubled to 25% in GS-infested areas. The direct physiological effects of GS likely played a decisive role in the death of wild apricots. Severe ductal embolism occurs in the xylem of wild apricots showing decline, which induces hydraulic dysfunction and reduces the synthesis and storage of non-structural carbohydrates [30]. Severe carbohydrate–water imbalance does not seem to defend against high-density GS. We expect that the high density of GS may continue to accelerate the death of wild apricots.

### 4.2. The Effect of GS on the Growth of Wild Apricots

Infested wild apricots had longer spring shoot lengths compared to non-infested trees (Figure 7). Trees with higher damage rankings had spring shoots that were approximately 2–3 times longer than those of non-infested trees. This a compensatory growth response to GS damage in wild apricots. Compensatory growth, such as increases in leaf area, new shoots, and root-to-crown ratio, is a positive response of plants to insect damage by altering resource allocation, physiology, and/or phenology [31,32,33]. Trees can be affected by increased sink demand from sap-sucking pests, inducing nutrients to move toward the damaged site [34]. The relocation of nutrients driven by GS feeding accumulated in fresh branches and led to increased shoot growth in infested branches (Figure 7). However, the health of trees gradually declined with time because of the accumulated damage and fewer sources of carbohydrates, or “carbon starvation” [30]. This means that there were insufficient nutrients allocated to the growth of branch, and the growth of spring shoots gradually decreased with time.

### 4.3. The Effect of GS on the Reproductive Capacity of Wild Apricots

Sap-feeding insects can significantly impair the reproductive capacity of woody plants [34,35]. This occurs primarily due to competition for vital nutrients between the sucking pests and the plant’s reproductive organs, such as buds, flowers, and fruits [34,36,37]. Previous research demonstrated a significant reduction in the fruit size of wild apricots infested with GS [18]. Our findings confirmed these observations, revealing significant declines in flower bud density, flowering percentage, and fruit yield in wild apricots with increasing levels of GS infestation. Furthermore, the detrimental impact of GS on wild apricot fruit production is exacerbated by the development of sooty mold, a secondary consequence of the infestation. Sooty mold has been shown to reduce stomatal conductance and photosynthetic capacity, ultimately leading to decreases in both the quantity and quality of fruit yield [13]. It is crucial to note that the flowering and fruit set of wild apricots are inherently susceptible to spring frosts, further compounding the challenges faced by these plants. In 2022, wild apricot yields reached 8 kg per plant in uninfested areas but was less than 1 kg per tree due to freezing conditions in 2023 and 2024 (Figure 9). The reduced yields due to GS and freezing will decrease the seed bank in soil, reducing recruitment of seedlings and ultimately forest regeneration and forest resilience [38].

### 4.4. Integrated Management Approach to GS

The changes in damage ranking, flowering status, and pest density were useful indexes for representing the ongoing recovery or damage in apricot forests in our study. These indexes can be employed for understanding the resilience and efficacy of IPM on GS and forest management. The implementation of integrated pest management (IPM) strategies can effectively reduce GS populations and help recover wild apricot forest health [18]. Large-scale aerial spraying programs have been successfully employed in wild apricot forests, resulting in significant reductions in GS densities across most areas [19]. Based on the GS population density results from this study, the densities of 1^st^- and 2^nd^-instar nymphs remain alarmingly high. Therefore, we recommend implementing comprehensive aerial spraying campaigns across wild apricot forests for effective control. In severely affected areas, targeted ground spraying should be additionally applied to achieve optimal pest management efficacy. Additionally, the pruning of infested branches has proven to be an effective approach for controlling GS populations and mitigating damage levels in wild apricot trees [39], as was the case with *Agrilus mali* in wild apples [40]. Our study observed a notable decline in the heavily damaged ranking during 2024, supporting the adoption of sustained pruning practices as a viable long-term strategy for mitigating damage in wild apricot forests. In the long term, biological control strategies should be prioritized as a sustainable and practical approach. This is supported by successful examples in Turkey, where parasitism rates from eight hymenopterous species for GS exceeded 70% in cultivated orchards [16]. The significantly reduced flowering rates and fruit yields raise concerns about the future regeneration of wild apricot populations. Furthermore, a comprehensive suite of nursery planting options to replace dead trees should be integrated with other management tactics to enhance the resilience of wild apricot forests and boost their ecological defenses against GS. For example, the postponement of spring grazing and reduction of grazing frequency can contribute to the survival of wild apricot seedlings and increase the stability of wild apricot tree populations.

## 5. Conclusions

Our study demonstrated that the population dynamics of GS, along with the damage rankings of wild apricots, have been gradually decreasing over the years. However, wild apricots still suffer damage because of the continuing high densities of GS. More than 25% of wild apricots died in the GS-infested areas. Furthermore, the cumulative damage inflicted by GS on wild apricots significantly stimulates the growth of new shoots but adversely affects their reproduction capacity, including the fruit yield, flowering percentage, and flower bud density. This cumulative damage effect becomes particularly pronounced as the damage ranking increases, raising concerns about the destiny of wild apricots. Moving forward, it is essential to strengthen the monitoring and integrated pest management (IPM) strategies against GS to first reduce its density below the damage threshold level. Additionally, these efforts should be integrated with other conservation and restoration strategies to enhance the resilience of wild apricots against GS by enhancing the recruitment of young trees in various landscapes in the Tianshan Mountains.

## Figures and Tables

**Figure 1 insects-16-00409-f001:**
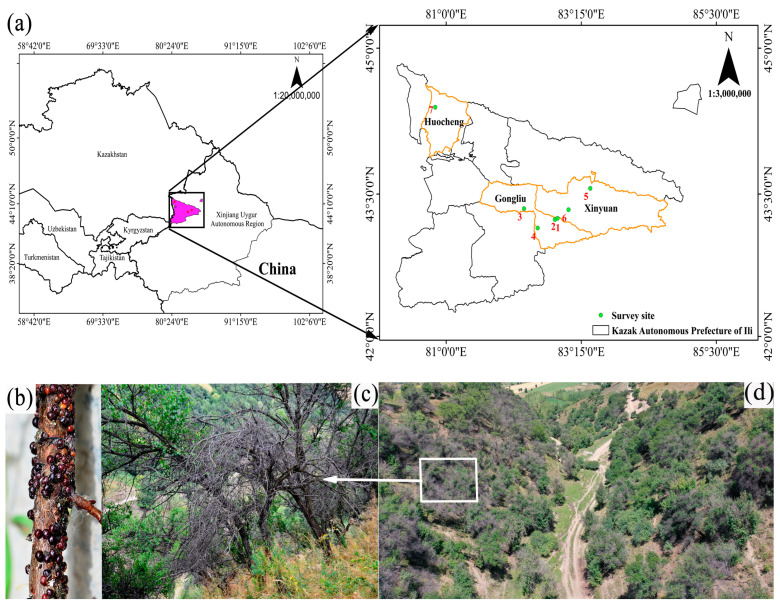
Locations of survey sites and damage status of wild apricot forests infested by *Sphaerolecanium prunastri* in Gongliu, Xinyuan, and Huocheng Counties, Xinjiang Uyghur Autonomous Region, China. (**a**) The dots in the map on the right indicate the 7 survey sites in the Yili River Valley: 1: Kuolesai; 2: Jinqikesai; 3: Keersenbulake; 4: Xiaomohuer; 5: Tuergen; 6: Talede; 7: Xiaoxigou. (**b**) The female scale insects adhere to the surface of the branches. (**c**) Desiccated and dead wild apricot branches resulting from infestation. (**d**) Regional damage status of wild apricot forests. Tree crowns that appear gray (arrow) have suffered severe damage.

**Figure 2 insects-16-00409-f002:**
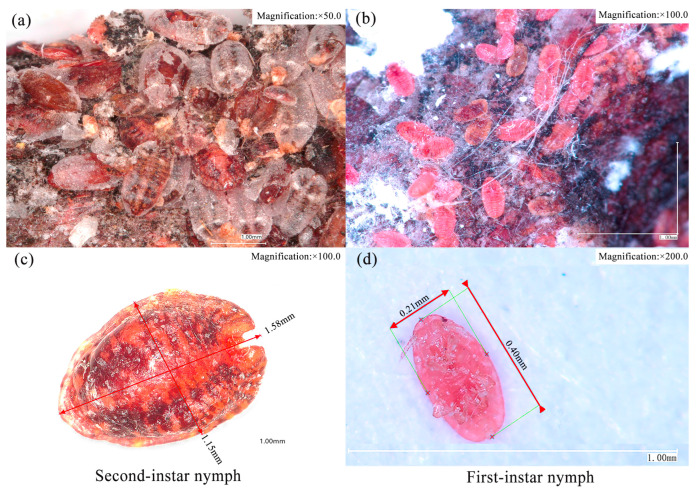
Morphological characteristics of first- and second-instar nymphs. (**a**) The morphology of 2^nd^-instar nymphs at 50× magnification; (**b**) The morphology of 1^st^-instar nymphs at 100× magnification; (**c**) The morphological size of a 2^nd^-instar nymph at 100× magnification; (**d**) The morphological size of a 1^st^-instar nymph at 200× magnification.

**Figure 3 insects-16-00409-f003:**
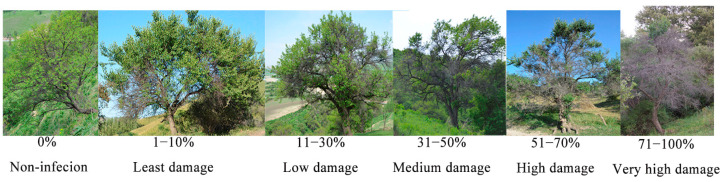
The damage rankings of wild apricot trees based on the percentage of branches damaged by GS in each tree.

**Figure 4 insects-16-00409-f004:**
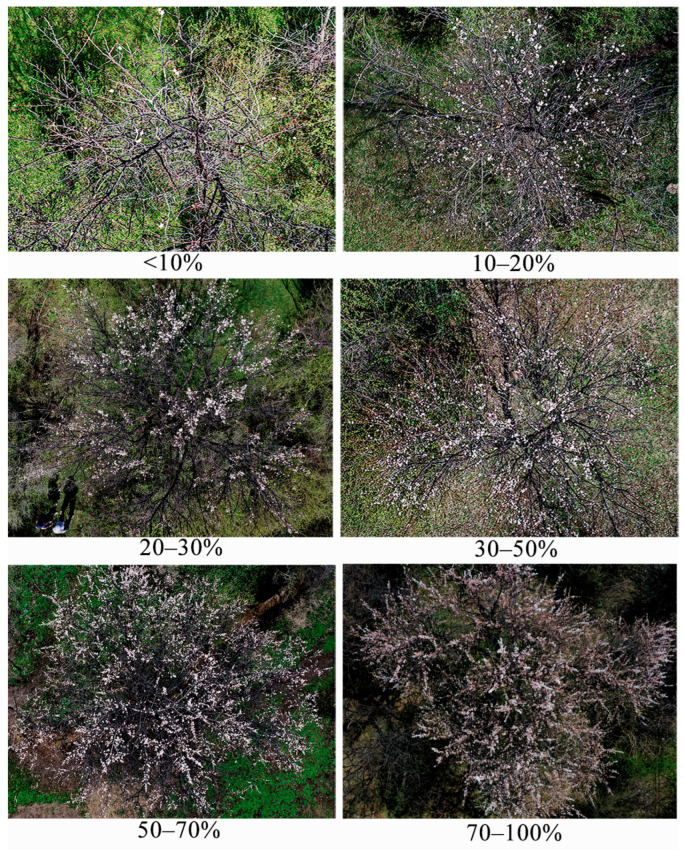
The flowering rate of wild apricot trees based on the percentage of blooms relative to the entire canopy.

**Figure 5 insects-16-00409-f005:**
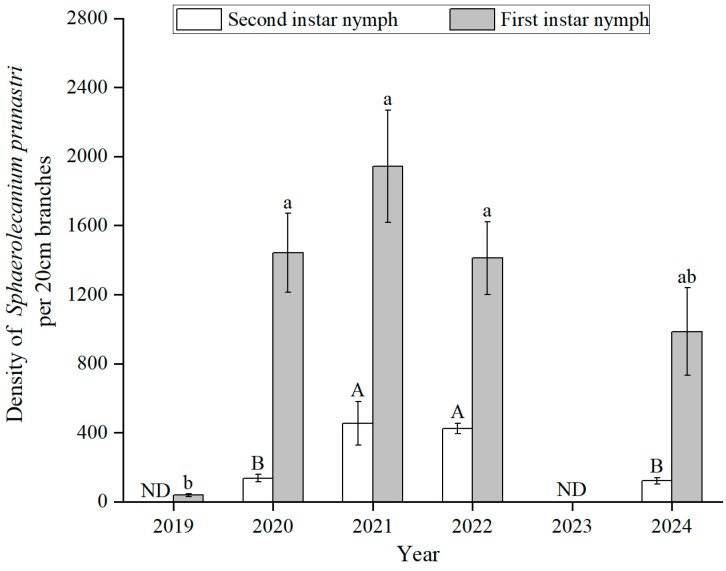
The density dynamics of 1^s^t- and 2^nd^-instar nymphs from 2019 to 2024 in wild apricot forests (no data in 2023). Capital letters represent a difference between 2^nd^-instar nymphs among years; lowercase letters represent a difference between 1^st^-instar nymphs among years. ND: no data.

**Figure 6 insects-16-00409-f006:**
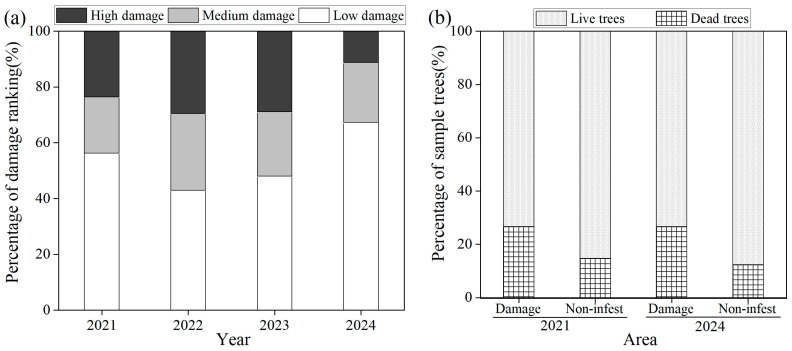
Proportion of trees with different damage rankings (**a**) and proportions of dead and live wild apricot trees over time (**b**).

**Figure 7 insects-16-00409-f007:**
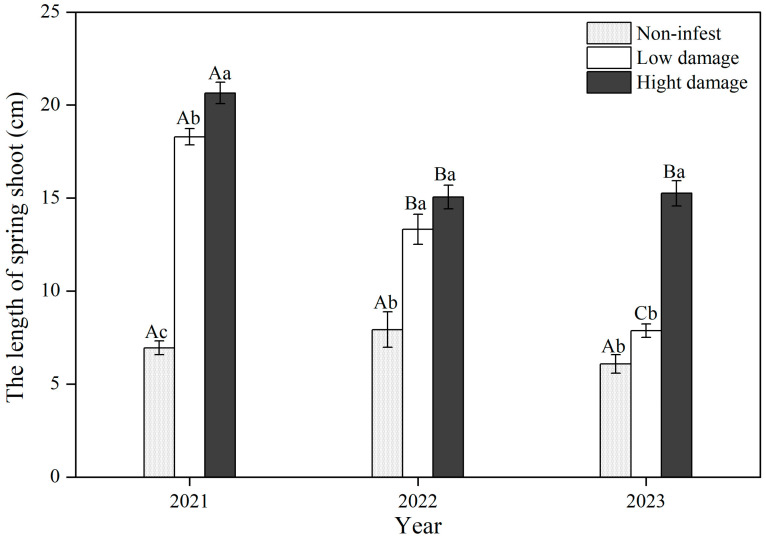
The lengths of spring shoots among three damage rankings. Capital letters represent a difference between years; lowercase letters represent a difference in the spring shoot length between the three damage rankings.

**Figure 8 insects-16-00409-f008:**
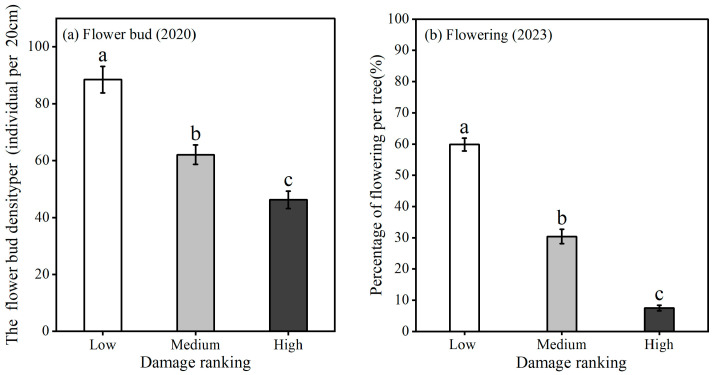
The density of flower buds (**a**) and percentage of flowering per wild apricot tree (**b**) associated with damage ranking. Lowercase letters represent a difference between the three damage rankings.

**Figure 9 insects-16-00409-f009:**
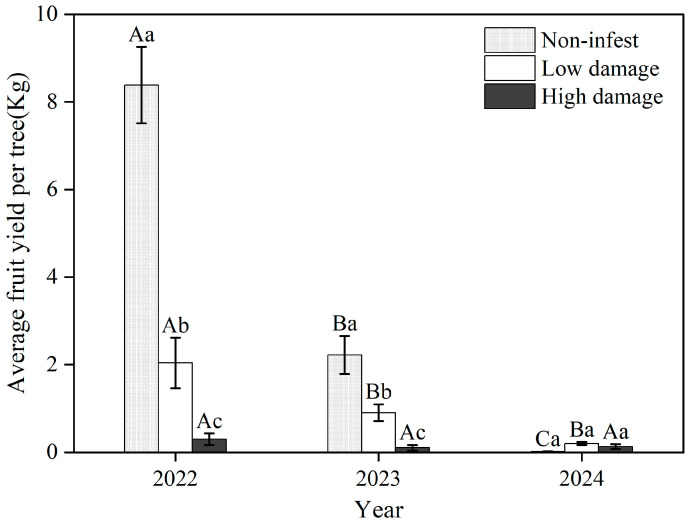
The fruit yield of wild apricots among three damage rankings. Capital letters represent a difference in fruit yield between years; lowercase letters represent a difference in fruit yield between the three damage rankings.

**Figure 10 insects-16-00409-f010:**
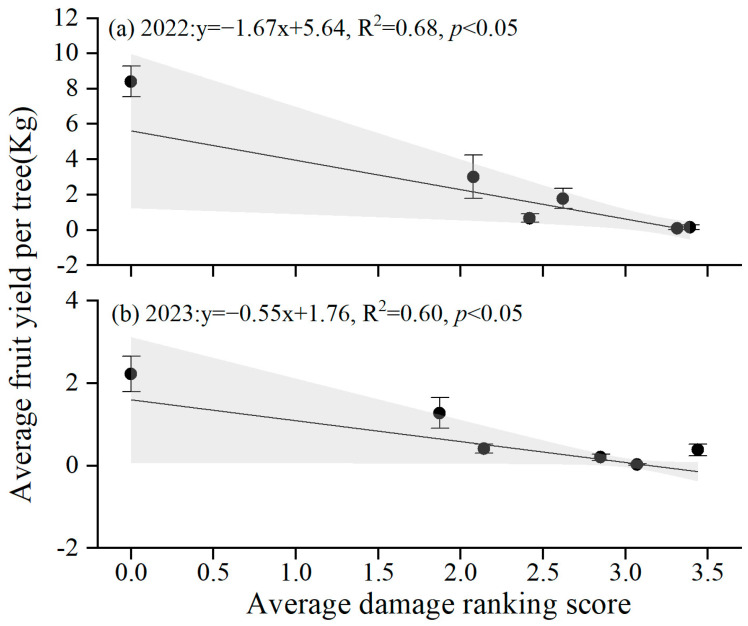
Average fruit yield per survey site relative to damage ranking score of wild apricots in 2022 (**a**) and 2023 (**b**). The line shows the linear regression.

**Table 1 insects-16-00409-t001:** Location of sample plots in remnant wild apricot forests in the Ili River Basin, Tianshan mountains. Tree density was assessed in 2021.

County	Survey Site	Plot Number	Latitude (N)	Longitude (E)	Altitude (m)	Height (m)	Diameter at Breast Height (cm)	Wild Apricot Tree Density (Number/ha)
Gongliu	Kuolesai	1	43.234	82.811	1248.10	7.93 ± 0.35	21.81 ± 1.48	224
		2	43.259	82.857	1343.23	7.28 ± 0.40	13.57 ±1.68	248
		3	43.248	82.697	1190.32	8.28 ± 0.24	23.78 ± 1.73	264
	Jinqikesai	1	43.248	82.857	1321.14	7.79 ± 0.37	22.2 ± 2.52	288
		2	43.246	82.857	1343.23	7.94 ± 0.43	21.96 ± 1.5	240
		3	43.248	82.858	1319.34	7.25 ± 0.44	15.57 ± 1.56	312
	Keersenbu lake	1	43.363	82.120	1252.33	6.30 ± 0.41	18.26 ± 1.36	264
		2	43.349	82.298	1237.30	8.31 ± 0.51	17.50 ± 1.71	232
		3	43.349	82.299	1211.59	8.05 ± 0.70	15.92 ± 0.98	240
	Xiaomohuer	1	43.178	82.733	1376.60	8.02 ± 0.26	27.99 ± 0.88	176
		2	43.141	82.442	1245.00	9.49 ± 0.45	22.25 ± 1.29	264
		3	43.177	82.733	1356.96	6.33 ± 0.57	27.52 ± 1.35	152
Xinyuan	Tuergen	1	43.561	83.482	1026.33	6.05 ± 0.56	19.98 ± 2.02	360
		2	43.560	83.401	1110.72	6.24 ± 0.61	26.6 ± 3.37	288
		3	43.539	83.485	1055.23	7.54 ± 0.95	19.92 ± 2.90	264
	Talede	4	43.338	83.039	1165.06	6.77 ± 0.51	20.94 ± 1.37	240
Huocheng	Xiaoxigou	1	44.380	80.818	1084.52	5.72 ± 0.47	27.17 ± 1.74	208
		2	44.429	80.832	1160.95	5.62 ± 0.22	25.37 ± 1.70	216
		3	44.394	80.814	1051.36	6.39 ± 0.22	18.57 ± 1.25	232

## Data Availability

The original contributions presented in the study are included in the article. Further inquiries can be directed to the corresponding authors.

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
