# Peer review of "The Post-Invasion Population Dynamics and Damage Caused by Globose Scale in Central Eurasia: Destiny of Wild Apricot Still at Stake"

_insects, 2025, doi:10.3390/insects16040409_

Round 1
Reviewer 1 Report
Comments and Suggestions for Authors
Sphaerolecanium prunastri is an important invasive pest that has caused significant harm to ecosystems, particularly in the Xinjiang region of China, where its impact on wild apricot trees is especially pronounced. Therefore, monitoring and research on this pest in the Xinjiang region is of great importance. Timely understanding of its spread and damage levels is crucial for developing effective control measures.
Review Comments:
- Title is too long: It is recommended to shorten and revise the title to make it more concise and clear.
- Lines 57-61: Four lines are used to describe a single example, making the explanation overly lengthy. It is suggested to condense this section and include two or more additional examples to strengthen the argument.
- Line 63: Please specify which country is the area of invasion for this pest.
- Lack of basic information: The paper is missing ecological photos of the pest, images depicting the extent of damage, descriptions of morphological features, and information about the pest's invasion range and severity in China. It is recommended to include these basic details.
- Lines 120-122: It is suggested to include a map of the sampling sites to help readers better understand the specific locations of the study areas.
- Table 1: The full form of "DBH" is not found in the main text. Please provide the full form.
- Lines 137-138: How can sampling objectivity be ensured? Some branches are severely damaged, while others may be less affected. Is there a possibility of bias toward sampling heavily damaged branches? How can sampling bias between different sites and different collectors be avoided?
- Lines 261-263: The sample in the figure includes only six data points. How can we ensure that the small sample size does not lead to overfitting and a spurious significance? Generally, with a small sample size, p-values are more likely to appear significant. Further discussion on this issue is recommended.
- Lines 327-329: If there is a “freezing” effect, how can the effects of freezing be distinguished from those caused by the pest? More details are needed for clarification.
- Discussion, first paragraph: The first paragraph contains excessive description of results. In the discussion section, it is not recommended to describe results in such detail in a single paragraph. It is suggested to condense this part.
- Redundancy between the first and second paragraphs of the discussion: There is some repetition regarding nymph density in the first and second paragraphs of the discussion. It is recommended to delete the first paragraph. While it summarizes the content of the later paragraphs, it seems unnecessary.
Author Response
Review Comments:
Title is too long: It is recommended to shorten and revise the title to make it more concise and clear.
Response: Thank you. We have revised the title to make it more concise and clear. The new title is: Destiny of wild apricot at stake: The post-invasion population dynamics of globose scale and forest damage in central Eurasia.
Lines 57-61: Four lines are used to describe a single example, making the explanation overly lengthy. It is suggested to condense this section and include two or more additional examples to strengthen the argument.
Response: We have condensed the original example and added another example to illustrate the severe impact of scale insect infestation (Line 60-63).
Line 63: Please specify which country is the area of invasion for this pest.
Response: We have added the specific information. It is specifically found in Spain, France, Italy, Belgium, the United States, Turkey, South Korea, Uzbekistan, Turkmenistan, and China (Line 68-69).
Lack of basic information: The paper is missing ecological photos of the pest, images depicting the extent of damage, descriptions of morphological features, and information about the pest's invasion range and severity in China. It is recommended to include these basic details.
Response: We have added the specific information in the Materials and Methods section. The ecological photographs of GS and the images of its damage in wild apricot forests have been added into the geographical location map of the sampling sites (Figure1). Furthermore, corresponding photographs have been added to the methodology for investigating 1st and 2nd instar GS (Figure2). The damage assessment criteria for wild apricots have been supplemented with photographic representations of each damage ranking (Figure3). Moreover, the photo of flowering rate of wild apricot tree is presented in the manuscript (Figure4).
Regarding the invasion and severe damage caused by this pest in China, its distribution has only been recorded in Shandong, Liaoning, Hebei, and Shaanxi provinces. However, there are no detailed reports on the extent of damage in these regions. In 2019, an outbreak was detected in Xinjiang, which has now become the most severely affected area. We have incorporated this information into the introduction (Line 71-73). We are pleased you suggested adding photos.
Lines 120-122: It is suggested to include a map of the sampling sites to help readers better understand the specific locations of the study areas.
Response: We have added a map of sampling sites in new figure 1.
Table 1: The full form of "DBH" is not found in the main text. Please provide the full form.
Response: We have corrected this.
Lines 137-138: How can sampling objectivity be ensured? Some branches are severely damaged, while others may be less affected. Is there a possibility of bias toward sampling heavily damaged branches? How can sampling bias between different sites and different collectors be avoided?
Response: Indeed, some branches on the sample trees were severely damaged, while others were less affected. However, the sample branches were randomly selected from the east, south, west, and north sides or angles in same height in each sampled tree (4-6 meters) in all plots (19) in different sites (7), covering varying degrees of damage and forest stands—thus avoiding oversampling of heavily infested branches.
In addition, the randomized sampling followed a systematic zigzag pattern: within each survey site, we traversed in a predefined Z-shaped transect and randomly selected sample trees until reaching the target count of 15 trees. This method with random sampling also help to mitigate the bias.
These details have been added to the density survey methodology section (Line 151-153).
Lines 261-263: The sample in the figure includes only six data points. How can we ensure that the small sample size does not lead to overfitting and a spurious significance? Generally, with a small sample size, p-values are more likely to appear significant. Further discussion on this issue is recommended.
Response: We do not make a great deal of the relationship as it is only based on 6 sites but combined with all our other observations (reduced flowering etc) it does suggest that yield reduction and average damage level are negatively related.
Lines 327-329: If there is a “freezing” effect, how can the effects of freezing be distinguished from those caused by the pest? More details are needed for clarification.
Response: The damage caused by freezing is easily identifiable. If freezing events happened in some years, flower damages were fully entirely caused by spring frost in spring. All the flowers suffered frost damage, showing brown necrotic symptoms, leading to a significant reduction in blooming. GS does not directly damage the flowers but only have indirect effects. When there is no freezing damage, the flower color remains normal. Here, we only indirectly assessed the long-term cumulative impact of scale insects on flowering. The freezing event affected all apricot population in that year. But the tree damage condition is easily distinguished while leave regrow in summer.
Discussion, first paragraph: The first paragraph contains excessive description of results. In the discussion section, it is not recommended to describe results in such detail in a single paragraph. It is suggested to condense this part.
Response: Thank you for your suggestions. We have streamlined and condensed the first paragraph as recommended (Line 303-310).
Redundancy between the first and second paragraphs of the discussion: There is some repetition regarding nymph density in the first and second paragraphs of the discussion. It is recommended to delete the first paragraph. While it summarizes the content of the later paragraphs, it seems unnecessary.
Response: We would still like to retain the first paragraph as an overall summary for the discussion section. Following your suggestion, we have modified the redundant information.
Reviewer 2 Report
Comments and Suggestions for Authors
The MS “Destiny of wild apricot (Armeniaca vulgaris) at stake: The post 2 invasion population dynamics of globose scaleSphaerolecanium prunastri and damage status of forests in central Eurasia” assessed the assessed globose scale (GS) population dynamics post invasion and its effects on growth and reproductive traits of wild apricot trees from 2019 to 2024 in Xinjiang, China. My biggest concern with the MS is that the authors attributed all differences in apricot growth, flower buds, and yield to the GS when there can be additional factors including diseases, other insects which may confound the results. These are good observations, but authors did not explain much about how these observations can be useful in devising management strategies for this insect. Additional observations are as below:
Methods and materials are well described. However, there are few questions those can be addressed to improve the MS. For example, authors randomly sampled 15 trees to assess nymph density. Did 15 sampled trees include any trees on the edge. If yes, how did authors eliminate the potential edge effect. How did authors confirm if the damage was from the GS and not from other biotic or abiotic factors such as a disease, other pest infestation, etc. Authors mention that control efficacy has been challenging due to the mountainous topography, adverse weather. How did these factors impact the natural mortality of trees.
Results: No significant differences were observed for first instar nymphs among 2020, 2021, 2022, and 2024. Authors should explain why they did not see increase in GS populations over the years. Were there any other factors that limited GS population increase? Authors mention control program was initiated in 2022, how did this impact the population increase or decline.
Discussion: Authors mention that a rapid increase in nymph density was observed from 2019 to 2021, culminating in a peak, followed by a gradual decline from 2022 to 2024. However, results section states that no significant differences were observed between the years. Authors also mention GS populations increased over years but a gradual decrease in damage levels within the wild apricot forests was observed. This is counter intuitive. Authors should explain the potential causes of decline in damage with increase in population. The discussion section about integrated management approach to GS mentions the success of biological control, aerial application etc. This contrasts with the introduction section which mentions that biological control and pesticide applications have limited success.
Author Response
The MS “Destiny of wild apricot (Armeniaca vulgaris) at stake: The post 2 invasion population dynamics of globose scaleSphaerolecanium prunastri and damage status of forests in central Eurasia” assessed the assessed globose scale (GS) population dynamics post invasion and its effects on growth and reproductive traits of wild apricot trees from 2019 to 2024 in Xinjiang, China. My biggest concern with the MS is that the authors attributed all differences in apricot growth, flower buds, and yield to the GS when there can be additional factors including diseases, other insects which may confound the results. These are good observations, but authors did not explain much about how these observations can be useful in devising management strategies for this insect. Additional observations are as below:
Response: Thank you for your valuable comments. Regarding your concern about potential influences from other pests and diseases on the observed apricot traits, our long-term monitoring (2019-2024) confirms that GS remains the primary pest affecting wild apricots, with no other significant pest impacts detected. The observed foliar diseases have shown minimal effects on tree growth based on other studies (as detailed in our response to specific comments).
Regarding the management implications, we have incorporated this discussion in Section 4.4 (Lines 362-364, 368-373, and 375-378) to better clarify the link between monitoring indicators and management strategies.
Methods and materials are well described. However, there are few questions those can be addressed to improve the MS. For example, authors randomly sampled 15 trees to assess nymph density. Did 15 sampled trees include any trees on the edge. If yes, how did authors eliminate the potential edge effect. How did authors confirm if the damage was from the GS and not from other biotic or abiotic factors such as a disease, other pest infestation, etc. Authors mention that control efficacy has been challenging due to the mountainous topography, adverse weather. How did these factors impact the natural mortality of trees.
Response: Firstly, regarding the survey of insect density. Survey sites were located within the forest and so had trees all around them. The 15 wild apricot trees were located within of the survey site, so we believe edge effects would not be an issue. This information has been added to line 151.
Secondly, concerning the question of how to determine that the damage was caused by GS rather than other biotic or abiotic factors, it is important to note that the density of GS in this area is extremely high, and it has rapidly erupted into a major pest in recent years, becoming a primary factor causing tree damage. A study has already shown that the mortality of wild apricot trees accelerates after GS infestation. Additionally, no other pests capable of causing significant damage have been observed in the wild apricot forests. As for disease-related damage, there is indeed a bacterial shot hole disease present in the wild apricot forests, but its occurrence is relatively low and mostly damages leaves. Compared to the direct damage caused by GS feeding on the branches of wild apricot trees, its impact on tree survival is minimal.
Furthermore, studies have already assessed that the impact of biotic factors significantly outweighs that of abiotic factors in wild apricot forests, except of course late spring freezing conditions which we observed and discuss. Moreover, our systematic four-year survey across multiple sites consistently confirmed that GS is the primary stressor affecting tree health.
Results: No significant differences were observed for first instar nymphs among 2020, 2021, 2022, and 2024. Authors should explain why they did not see increase in GS populations over the years. Were there any other factors that limited GS population increase? Authors mention control program was initiated in 2022, how did this impact the population increase or decline.
Response: the first instar nymph population did not decline but showed a downward trend by 2024 and second instars had indeed declined by 2024. This has been noted in the manuscript. In fact, localized management practices such as pruning and drone-based pesticide application had already been initiated prior to the large-scale control program launched in 2022. But despite these initial measures populations continued to increase until 2021. While native natural enemies were unable to reduce the pest population density, they likely exerted some suppressive effect. Furthermore, density-dependent factors, such as habitat limitation at high population levels may have constrained further population growth. The 2022 control program contributed to population reduction, though its effectiveness appears to have been limited. Unlike orchards, this is a forest situation and variation amongst sites and trees is high in part due to the topography and the other surrounding trees. Moreover, we found no clear relationship between first and second instar nymphs, particularly overwintering second nymphs which is more detrimental to tree health.
Discussion: Authors mention that a rapid increase in nymph density was observed from 2019 to 2021, culminating in a peak, followed by a gradual decline from 2022 to 2024. However, results section states that no significant differences were observed between the years. Authors also mention GS populations increased over years but a gradual decrease in damage levels within the wild apricot forests was observed. This is counter intuitive. Authors should explain the potential causes of decline in damage with increase in population. The discussion section about integrated management approach to GS mentions the success of biological control, aerial application etc. This contrasts with the introduction section which mentions that biological control and pesticide applications have limited success.
Response: First, we sincerely apologize for the misunderstanding. In fact, the observed increase in GS population density does not conflict with the recorded decline in damage severity by 2024. Our results clearly indicate GS population increased from 2019, first instars have stayed stubbornly high but appear to declining by 2024 and second instars have declined. Damage assessments are from 2021 and some measures from 2022 as we clearly state, and that has trended down.
In addition, the integrated pest management (IPM) approach involving GS is consistent with the control methods in the Introduction section. Although biological control and aircraft spraying are mentioned in this article as limited, this is only for the expectation that the pest population can be reduced on a large scale, and this does not mean that the control of the pest is ineffective. We suggest as it is an invasive a classical biological control program needs to be considered
Round 2
Reviewer 2 Report
Comments and Suggestions for Authors
The MS has been significantly improved.